# Hydration and Ion Pair Formation in Aqueous Lu^3+^- Solution

**DOI:** 10.3390/molecules23123237

**Published:** 2018-12-07

**Authors:** Wolfram Rudolph, Gert Irmer

**Affiliations:** 1Medizinische Fakultät der TU Dresden, Institut für Virologie im MTZ, Fiedlerstr. 42, 01307 Dresden, Germany; 2Technische Universität Bergakademie Freiberg, Institut für Theoretische Physik, Leipziger Str. 23, 09596 Freiberg, Germany; irmer@physik.tu-freiberg.de

**Keywords:** Raman spectroscopy, Lu(ClO_4_)_3_, Lu(CF_3_SO_3_)_3_ and LuCl_3_ solutions in H_2_O and D_2_O, Lu^3+^- hydration, Lu-O skeleton modes, monochloro complex of Lu^3+^, DFT calculations of [Lu(OH_2_)_8_]^3+^, polarizable dielectric continuum model

## Abstract

Aqueous solutions of Lu^3+^- perchlorate, triflate and chloride were measured by Raman spectroscopy. A weak, isotropic mode at 396 cm^−1^ (full width at half height (fwhh) at 50 cm^−1^) was observed in perchlorate and triflate solutions. This mode was assigned to the totally symmetric stretching mode of [Lu(OH_2_)_8_]^3+^, ν_1_LuO_8_. In Lu(ClO_4_)_3_ solutions in heavy water, the ν_1_LuO_8_ symmetric stretch of [Lu(OD_2_)_8_]^3+^ appears at 376.5 cm^−1^. The shift confirms the theoretical isotopic effect of this mode. In the anisotropic scattering of aqueous Lu(ClO_4_)_3_, five bands of very low intensity were observed at 113 cm^−1^, 161.6 cm^−1^, 231 cm^−1^, 261.3 cm^−1^ and 344 cm^−1^. In LuCl_3_ (aq) solutions measured over a concentration range from 0.105–3.199 mol·L^−1^ a 1:1 chloro-complex was detected. Its equilibrium concentration, however, disappeared rapidly with dilution and vanished at a concentration < 0.5 mol·L^−1^. Quantitative Raman spectroscopy allowed the detection of the fractions of [Lu(OH_2_)_8_]^3+^, the fully hydrated species and the mono-chloro complex, [Lu(OH_2_)_7_Cl]^2+^. In a ternary LuCl_3_/HCl solution, a mixtrure of chloro-complex species of the type [Lu(OH_2_)_8−n_Cl_n_]^+3−n^ (*n* = 1 and 2) were detected. DFT geometry optimization and frequency calculations are reported for Lu^3+^- water cluster in vacuo and with a polarizable dielectric continuum (PC) model including the bulk solvent implicitly. The bond distance and angle for [Lu(OH_2_)_8_]^3+^ within the PC are in good agreement with data from structural experiments. The DFT frequencies for the Lu-O modes of [Lu(OH_2_)_8_]^3+^ and its deuterated analog [Lu(OD_2_)_8_]^3+^ in a PC are in fair agreement with the experimental ones. The calculated hydration enthalpy of Lu^3+^ (aq) is slightly lower than the experimental value.

## 1. Introduction

The element lutetium (Lu), a silvery white metal with the atomic number 71, is the heaviest of the lanthanides and has a full f-electron shell with a configuration: [Xe] 4f^14^5d^1^6s^2^. With a crustal abundance of 0.32 ppm [1,2], it is one of the rarest of the lanthanides. Because of this rarity and accompanying high price, only a few applications of this metal are known to date. Lutetium is used as a catalyst for cracking hydrocarbons in oil refineries [3,4]. Lutetium aluminum garnet (LuAG, Lu_3_Al_5_O_12_) is primarily known for its use in high-efficiency laser devices [5]. Lutetium oxyorthosilicat (LSO), a silicon oxygen compound of lutetium doped with cerium, is utilized as a detector for positron emission tomography [6]. Naturally occurring lutetium comprises two isotopes: ^175^Lu and ^176^Lu. ^176^Lu is unstable with an extremely long half-life of 3.78 × 10^10^ years which makes it suitable for determining the age of meteorites [7]. A short lived radionuclide ^177^Lu (half-life of 6.6430 days [8]) is used in the treatment of certain tumors, mainly neuroendocrine tumors [9].

In aqueous solution, lutetium exists exclusively in the trivalent state, Lu^3+^, and is strongly hydrated due to its high charge to radius ratio. Trivalent lutetium has the smallest ionic radius of all 15 lanthanides (Ln^3+^) across the series from lanthanum to lutetium known as the lanthanide contraction. Lu^3+^ hydrolyses most, i.e. shows a lesser basicity, than the preceding trivalent rare earth ions in aqueous solution. The difference in basicity is the basis for all separation methods of the individual rare earth ions [10]. The hydrolysis constant for the first monomeric species, β_1_^o^ at 25 °C for LuOH^2+^ is 4.677 × 10^−8^, the largest across the series, while for LaOH^2+^, the lowest, it is 1.288 × 10^−9^ [11]. However, compared to other trivalent metal ions such as Al^3+^ or Fe^3+^, the hydrolysis is much less pronounced.

Light rare earth ions are nonahydrates while the heavy rare earth ions are octahydrates in aqueous solution with weakly complex forming anions. The crossover of the hydration number from 9 to 8 occurs in the middle of the series. This sudden change in hydration number, known as *gadolinium break*, was used to explain experimental data in aqueous solution [12,13,14,15]. Recent studies, however, concluded that no such break exists and that the hydration number of Ln^3+^ ions smoothly changes from 9 to 8 across the series [16,17,18] with the rare earth (RE) ions of the series having non-integer numbers between 9 and 8 in the middle of the series. This means that the water ligands in the nona-/octa- hydrates exchange rapidly with each another [19,20].

For the Lu^3+^ ion in aqueous solution, the local hydration structure was measured using extended X-ray absorption fine structure (EXAFS) spectroscopy [16,21,22,23,24] and X-ray diffraction [25,26]. The majority of studies agreed with the 8-fold coordination of water molecules grouped around Lu^3+^ in square antiprismatic (SAP) geometry. A combined study using molecular dynamics (MD) simulation and EXAFS measurement proved the SAP structure of the Lu^3+^- octahydrate and the Lu^3+^-O bond distance was given at 2.32 Å [21]. 

The reported Lu-O bond distances range from 2.282 Å to 2.35 Å. These variations may be due to the fact that XRD measured concentrated solutions while EXAFS used relatively dilute solutions [16,21]. However, such differences in concentration scale, pose a problem because ion pairs are likely to form in concentrated solutions [26,27,28,29]. Ion association studied on these solutions do not agree on the nature of the ion pairs/complex species formed. In addition to the detected complex species, there is no doubt that outer-sphere and outer-outer sphere ion pairs exist in rare earth solutions. A recent dielectric relaxation spectroscopy (DRS) study on chloride, nitrate and sulfate solutions of La^3+^ and Eu^3+^ claimed, exclusively, outer-sphere and outer-outer-sphere ion pairs in these solutions [30]. However, in a variety of studies applying a wide array of methods such as solvent extraction [31,32,33,34], ion exchange [35,36,37], fluorescence spectroscopy [38,39,40], ultrasound absorption [41,42,43,44] and NMR spectroscopy [45,46,47], it was convincingly shown that complex species also exist in aqueous solution contrary to the new DRS results [30] put forward. 

Raman spectroscopy probes the immediate environment of metal ions in solution and was frequently used to study hydrated metal ions and ion association. RE-ions, however, have only recently been measured at a sufficiently high quality in the low frequency region of the Raman spectrum. These spectra then allow the construction of the isotropic scattering profile from the measured data. A highly polarized band in the low frequency region of the Raman spectrum due to the symmetrical metal-oxygen mode of the hydrated cations is the most characteristic peak [26,27,28,29,48,49,50,51]. This mode is sensitive to possible ion pair formation. Occasionally, all the theoretically predicted bands of the metal-oxygen modes may be observed and used to support the assignment of the point group symmetry and the coordination number in these aqueous solutions. Raman scattering measurements on Lu^3+^ (aq) should allow, in principle, the characterization of the solution structure in greater detail. No systematic Raman studies on aqueous lutetium solutions exist but one study on a glassy LuCl_3_ (aq) sample at high concentrations was reported [52]. 

The present study was undertaken to characterize the hydration and speciation in aqueous Lu^3+^ solutions and to this end Lu^3+^-salt solutions with common anions (ClO_4_^−^, CF_3_SO_3_^−^ and Cl^−^) were considered over a broad concentration range and down to the low frequency region. Triflate (trifluoromethanesulfonate) and perchlorate are considered weakly-complex-forming anions and were, therefore, chosen to measure the Lu-O stretching modes in aqueous solution so as to identify and assign bands unique to the first hydration sphere of Lu^3+^ (aq). A Lu(ClO_4_)_3_ solution in heavy water was also measured to characterize the vibrational isotope effect by changing from [Lu(H_2_O)_8_]^3+^ to [Lu(D_2_O)_8_]^3+^. In a variety of di- and trivalent metal ion solutions with chloride as counterion, however, it has been shown that these anions readily form complexes [27,28,29] and the question arises as to whether chloro- complex species occur in Lu^3+^ (aq) solutions. A LuCl_3_ solution series and a ternary solution of LuCl_3_/HCl were studied in order to answer the question whether chloro-complex species exist and if possible to quantify the fraction of the chloro-complex species.

In addition, simulations on lutetium-water species with eight water molecules were considered by applying density functional theory (DFT) in the gas phase. Furthermore, the Lu^3+^- species with a polarizable dielectric continuum were also simulated in order to take into account the effects of the bulk solvent. Frequency calculations were carried out via the calculation of the second derivative of the energy with respect to the nuclear positions of the optimized Lu^3+^-water species. Furthermore, the hydration enthalpy of the Lu^3+^ ion at 25 °C was modeled.

## 2. Experimental Details; Data Analysis and DFT Calculations on Lu^3+^-Water Hydrates

### 2.1. Preparation of Solutions

Lutetium stock solutions were prepared and the lutetium concentrations of these solutions were analysed by complexometric titration [53]. The solution densities were determined with a pycnometer at 23 °C and the molar ratios of water per salt were calculated (R_w_-values). For Raman spectroscopic measurements, the solutions were filtered through a fine sintered glass frit (1–1.6 µm pore size). The solutions showed no Tyndall effect and were “optically empty” [54]. 

Lutetium perchlorate solutions were prepared from Lu_2_O_3_ (99.9%, Merck KGaA, Darmstadt, Germany) and HClO_4_ in a beaker until all oxide dissolved. A Lu(ClO_4_)_3_ stock solution was prepared at 2.233 mol·L^−1^ (R_w_ = 19.01). The slightly acidified solution using HClO_4_ had a pH value at ~1.0. From this stock solution, three dilute solutions were prepared: 1.132 mol·L^−1^ (R_w_ = 42.77), 0.556 (R_w_ = 92.40), and 0.186 mol·L^−1^ (R_w_ = 289.0). The solutions were analyzed for dissolved chloride with a 5% AgNO_3_ solution. The absence of a white AgCl precipitate was proof that the stock solution was free of Cl^−^.

Two Lu(ClO_4_)_3_ solutions in heavy water were prepared from a deuterated Lu(ClO_4_)_3_ stock solution and with 99.9 atom % D (Sigma-Aldrich) at 1.398 mol·L^−1^ and 0.439 mol·L^−1^. The deuteration degree of the 0.439 mol·L^−1^ Lu(ClO_4_)_3_ solution in D_2_O was better than 97% D.

A 1.05 mol·L^−1^ Lu(CF_3_SO_3_)_3_ solution prepared from anhydrous Lu(CF_3_SO_3_)_3_ (99.9%, Sigma-Aldrich,) and triply distilled water. The solution had a pH value at ~ 1.0.

A stock solution of LuCl_3_ at 3.199 mol·L^−1^ (R_w_ = 15.68) was prepared from Lu_2_O_3_ (99.9%, Sigma-Aldrich,) and HCl in a beaker covered with a glass lid until all oxide dissolved. The pH value of the stock solution was 0.85. The following solutions were prepared from the stock solution and triply distilled water by weight: 1.890 mol·L^−1^ (R_w_ = 28.51), 0.935 mol·L^−1^ (R_w_ = 59.00), 0.478 mol·L^−1^ (R_w_ = 115.99), 0.241 mol·L^−1^ (R_w_ = 229.86) and 0.105 mol·L^−1^ (R_w_ = 528.65). Furthermore, a solution with an excess of HCl (37%, zur Analyse, Merck KGaA, Darmstadt, Germany) was prepared from the LuCl_3_ stock solution and a 37% HCl solution (37%, zur Analyse, Merck, Darmstadt, Germany) by weight. The ternary solution contained 1.589 mol∙L^−1^ LuCl_3_ and 6.047 mol∙L^−1^ HCl. An HCl solution was also prepared at 6.04 mol·L^−1^ from a 37% HCl solution.

### 2.2. Spectroscopic Measurements:

Raman spectra were measured in the macro chamber of the T 64000 Raman spectrometer from Jobin Yvon in a 90° scattering geometry at 23 °C. These measurements have been described elsewhere [49,55]. A quartz cuvette from Hellma Analytics (Müllheim, Germany) with 10 mm path length and a volume 1000 µL was used. Briefly, the spectra were excited with the 487.987 line of an Ar^+^ laser at a power level of 1100 mW at the sample. After passing the spectrometer in subtractive mode, with gratings of 1800 grooves/mm, the scattered light was detected with a cooled CCD detector. The scattering geometries I_VV_ = (X[ZZ]Y) and I_VH_ = (X[ZX]Y) are defined as follows: the propagation (wave vector direction) of the exciting laser beam is in X direction and the propagation of the observed scattered light is in Y direction, the 90° geometry. The polarisation (electrical field vector) of the laser beam is fixed in Z direction (vertical) and the polarisation of the observed scattered light is observed in Z direction (vertical) for the I_VV_ scattering geometry. For I_VH_ the fixed electric field vector of the exciting laser beam in Z direction (vertical) and the observed scattering light is polarized in the X direction (horizontal). Thus, for the two scattering geometries it follows:I_VV_ = I(X[ZZ]Y) = 45α′^2^+ 4γ′^2^(1)
I_VH_ = I(X[ZX]Y) = 3γ′^2^(2)

The symboles ${\bar{\alpha }}^{\prime }$ and γ′ are the isotropic and the anisotropic invariant of the Raman polarizability tensor, respectively [49,55]. The isotropic spectrum, I_iso_ was constructed according to Equation (3): I_iso_ = I_VV_ − 4/3 × I_VH_(3)

The polarization degree of the Raman bands, ρ (ρ = I_VH_/I_VV_) was determined using an analyzer and adjusted, if necessary, before each measuring cycle using CCl_4_. A detailed account of this procedure may be found in ref [49,55].

In order to characterize the spectral features in the low wavenumber region, the Raman spectra in I-format were reduced and the R-spectra obtained. The R- spectra have been constructed according to the procedure previously described [49]. The I-spectra were corrected for the scattering factor (ν_L_-$\bar{v}$)^3^ and further corrected for the Bose-Einstein temperature factor, B = [1 − exp(−h$\bar{v}$c/kT)] and the frequency factor, $\bar{v}$, to give the so called reduced or R($\bar{v}$) spectrum. The isotropic spectrum in R-format arise from the corrected R_VV_ and R_VH_ spectra according to Equation (4):R (v)_iso_ = R(v)_VV_ − 4/3R(v)_VH_.(4)

In the low wavenumber region, the I($\bar{v}$) and R($\bar{v}$) spectra are significantly different and only the spectra in R-format are presented. It should be noted that one of the advantages of using isotropic R-spectra is that the baseline is almost flat in the 50–700 cm^−1^ wavenumber region allowing relatively unperturbed observation of the presence of any weak modes [49,51].

### 2.3. Quantitative Raman Measurements

The quantitative relationship between the band intensity and the solute concentration by Raman measurements is the simple equation: I_i_ = J_i_ × C_T_, where I_i_ is the integrated band intensity of band i of the molecular species as a function of the scattering coefficient J_i_ multiplied by the solute concentration, C_T_ in mol·L^−1^. The perchlorate band in these solutions served as the internal standard. The result of the functional relationship between the integrated band intensity of the ν_1_LuO_8_ mode in Lu(ClO_4_)_3_ (aq) and the concentration of Lu^3+^ (aq) is given by Equation (5): I_396_ = 1148.3 × C_T_(5)

The coefficient of determination, R^2^ is 0.99. The relationship holds if Lu(ClO_4_)_3_ (aq) is completely dissociated into its ions and C_T_ is equal to the concentration of Lu^3+^. The linearity of the integrated band intensity I_396_ versus solute concentration of Lu(ClO_4_)_3_ up to a concentration at ~3 mol·L^−1^ proves this point. The LuCl_3_ solutions were then measured under the same conditions as the Lu(ClO_4_)_3_ solutions and the integrated band intensities for the ν_1_LuO_8_ mode could be determined and compared to the one in the perchlorate solution. The accuracy of the band intensities in these solutions was estimated to be ± 5%. 

### 2.4. DFT Calculations

The calculations were executed using the Gaussian03 package [56] employing the B3LYP functional [57]. The Stuttgart/Dresden (SDD) basis set was used which adequately reproduces the geometrical parameters, in particular the experimentally observed Lu–O distance. SDD uses a relativistic effective core potential (ECP) for Lu^3+^. All electrons for the other atoms are described by Dunning/Huzinaga valence double-zeta (D95V) functions. Placing the octa-hydrate in a solvent continuum employing the Polarized Continuum Model (PCM) which takes into account the solvation effect of bulk water gave significantly better results compared with the experimental frequencies. The PCM used was the version described in ref. [58] where the solvent is modelled as an isotropic and homogeneous continuum, characterized by its dielectric properties. The cavity is defined as a set of interlocking spheres attached to the solute atoms. The electrostatic solute-solution interaction is calculated introducing an apparent charge distribution spread over the cavity surface.

Geometries of gas phase hydrates and within the PC framework with 8 water molecules surrounding the Lu^3+^ ion were optimized. The aqua ion with S_8_ symmetry was the only structure, which led to an energy minimum without imaginary frequencies. The geometrical data and the frequency of the breathing mode of the [Lu(H_2_O)_8_]^3+^ ion will be discussed below. Furthermore, the hydration enthalpy of Lu^3+^ was simulated applying the DFT method and this procedure is presented in App. A.

## 3. Results

### 3.1. The [Lu(OH_2_)_8_]^3+^ (aq) Ion

Lu^3+^ is strongly hydrated in aqueous solution as revealed by its large standard molar enthalpy of hydration (ΔH^ϴ^_hyd_) at −3640 kJ·mol^−1^. However, the ΔH^ϴ^_hyd_ —values reported in the literature scatter from −3530 to −3695 kJ·mol^−^^1^ [59,60,61]. Our DFT value for ΔH^ϴ^_hyd_ at −3847.7 kJ·mol^−1^ is slightly smaller than the thermodynamic value (calculation procedure see Appendix B). The DFT optimized [Lu(H_2_O)_8_]^3+^ geometry in the gas phase and also with a polarizable continuum (PC) model gave a square antiprismatic coordination polyhedron with symmetry S_8_ (Figure 1). Figure 1 shows both the hydrated ion and its LuO_8_ skeleton. Furthermore, the [Lu(H_2_O)_8_]^3+^ aqua ion and its deuterated analog, [Lu(D_2_O)_8_]^3+^, imbedded in a PC were successfully simulated. The calculated Lu-O bond distance of the [Lu(H_2_O)_8_]^3+^ ion with a solvation sphere (taking into account the influence of the bulk water) is in good agreement with the experimental value (Table 1). The discussion of the DFT frequencies of the cluster will be given below.

The vibrational analysis of the [Lu(OH_2_)_8_]^3+^ (S_8_ symmetry) with its 69 normal modes (n.m.s) leads to the irreducible representation: Γ_v_(S_8_) = 8a(Ra) + 9b(i.r.) + 18e_1_(Ra, i.r.) + 18e_2_(n.a.) + 16e_3_(n.a.) (Ra as Raman, i.r. as infrared and n.a. as not active). The modes with character a and e_1_ are Raman active and those with character b and e_3_ are i.r. allowed. The vibrations can be divided into 24 internal and 24 external vibrational modes of the coordinated water molecules plus 21 n.m.s of the LuO_8_ skeleton (a similar procedure has been given in ref. [62]). 

The internal and external vibrations of the coordinated water may be considered separate from those of the LuO_8_ skeleton with the ligated water molecules seen as point masses. The LuO_8_ skeleton possesses D_4d_ symmetry and with its 9 atoms leads to 21 n.m.s and the irreducible representation is as follows: Γ_v_(D_4d_) = 2a_1_(Ra) + b_1_(i.a.) + 2b_2_(i.r.) + 3e_1_(i.r.) + 3e_2_(Ra) + 2e_3_(Ra). Although the LuO_8_ skeleton possesses no symmetry centre, the mutual exclusion rule is nevertheless effective. Seven modes with the character a_1_, e_2_ and e_3_ are Raman active while six modes with the character b_1_, b_2_ and e_1_ are i.r. allowed. The symmetric Lu-O stretch, the breathing mode, is only Raman active and appears strongly polarized in the Raman spectrum as the strongest band of the LuO_8_ skeleton spectrum. Two additional depolarized Raman active stretching modes are expected (character e_2_ and e_3_) as well as four other Raman deformation modes (character a_1_, e_2_ and e_3_). In I.R., two stretching modes (character b_2_ and e_1_) are expected and the remaining are deformations. In reality, however, the skeleton modes may not always be easily detected because they appear quite broad, weak and even obscured by the water background in solution.

Considering the coordinated water molecules of the [Lu(OH_2_)_8_]^3+^- ion, the internal and external n.m.s of the water molecules may be divided into 24 internal and 24 external vibrations. These vibrations are derived from the rotational- and translational degrees of freedom of the isolated water molecule. The n.m.s of these water molecules, the external modes, are librational modes such as wag, twist, and rock [49,62]. In aqueous solutions, however, the librational modes are strongly overlapped with the librations of the bulk water and therefore not easily detected. They appear as weak, very broad modes below 1200 cm^−1^ [62]. In addition to these librational modes, internal water modes are observed, the deformation mode, ν_2_(H_2_O) and two stretching OH modes, ν_1_ and ν_3_. The deformation mode in liquid water is found at 1640 cm^−1^ and the stretching modes at ~3400 cm^−1^ appear as a very broad structured band of H-bonded water molecules. The water modes are modified when coordinated to metal ions such as Lu^3+^ but are difficult to separate from the contributions of the librational and internal water modes of the bulk phase. In neat liquid water, the H-bonded water molecules show broad and weak librational modes and internal water modes, the deformation band, δ H-O-H, and the stretching O-H bands [29,62]. Spectra of liquid water and heavy water, bands and band assignments are given elsewhere [62].

The hydration sphere of Lu^3+^ (aq) is somewhat labile and a water-exchange rate constant k_ex_ at 25 °C was given at 6 × 10^7^ s^−1^ (from H_2_O-SO_4_^2−^ interchange rates [42,43]) with a water residence time τ = 16.7 ns and is slightly more labile than for instance [Y(OH_2_)_8_]^3+^ (aq) with τ = 50 ns [29]. On the other hand, it has a more rigid hydration sphere than [La(OH_2_)_9_]^3+^ with τ = 5 ns (aq). [27]. (The vibrational duration for the Lu-O breathing mode is 0.084 ps, short enough to allow ~ 199,000 vibrations before the cluster experiences a water exchange. In other words, Raman spectroscopy probes an average hydration structure of rapidly exchanging water molecules in the direct vicinity of the Lu^3+^ ion.). 

### 3.2. Lu(ClO_4_)_3_ and Lu(CF_3_SO_3_)_3_ Solutions in Water and Heavy Water

Lu(ClO_4_)_3_ solution spectra: A Raman spectrum in the low frequency range of a 0.186 mol·L^−1^ Lu(ClO_4_)_3_ solution (water to salt ratio at 289.0) is presented in Figure 2. Additionally, Raman spectra of two more concentrated Lu(ClO_4_)_3_ solutions at 2.233 mol·L^−1^ (R_w_ = 19.01) and at 0.556 mol·L^−1^ (R_w_ = 92.37) are presented in Appendix A. The perchlorate ion has been chosen as a counterion because it is known as a weakly complex-forming ion. A high frequency band at 3542 cm^−1^ (fwhh = 90 cm^−1^), in the O-H stretching band region of H_2_O, is observed in Lu(ClO_4_)_3_ solutions which is attributed to an O-H band of weakly hydrated perchlorate ion (Appendix A). (In addition to the stretching mode ν(O-H···ClO_4_^−^) at 3542 cm^−1^ a very weak, broad mode appears in the terahertz region at ~165 cm^−1^ in Lu(ClO_4_)_3_ solutions (R_iso_). The latter mode has an equivalent in pure water at ~175 cm^−1^ where it is moderately intense and slightly polarized. It is the restricted translational mode of the H- bonded water molecules (O···O-H)). In concentrated Lu(ClO_4_)_3_ solutions other H-bonds are important, namely OH···OClO_3_^-^ and the intensity of the band due to HOH··· OClO_3_^−^ is extremely weak in the isotropic Raman spectrum [27,28,29]). For a detailed discussion on the influence of ClO_4_^−^ on the stretching band on water in, for instance, La(ClO_4_)_3_ (aq) and Ce(ClO_4_)_3_ (aq) see [27,28].

It is noteworthy to mention that the stretching band of O-H···OClO_3_^−^ is slightly cation dependent because of the different charge to radius ratios (polarizing power) of these ions. In Lu(ClO_4_)_3_ (aq), the band appears at 3542 cm^−1^, slightly lower than in La(ClO_4_)_3_ (aq) where it appears at 3550 cm^−1^ [27].

The Raman spectrum of ClO_4_^−^ (aq) (T_d_ symmetry) has been discussed in detail elsewhere so only a brief discussion shall be given [27,28,29]. The ClO_4_^−^ ion possesses nine vibrational degrees of freedom and its internal vibrations span the representation Γ_vib_(T_d_) = a_1_(Ra) + e(Ra) + 2f_2_(Ra, i.r.). All four n.m.s are Raman active, but in i.r. only the f_2_ modes are allowed. In dilute solution, the symmetric Cl-O stretch, ν_1_(a_1_) ClO_4_^−^ appears at 931.5 cm^−1^ and is totally polarized (ρ = 0.005) (fwhh = 7.1 cm^−1^) and is the strongest mode in the Raman spectrum. The antisymmetric stretch, ν_3_(f_2_) ClO_4_^−^ centred at 1105 cm^−1^ and the deformation modes ν_4_(f_2_) ClO_4_^−^ at 629 cm^−1^ and ν_2_(e) ClO_4_^−^ at 458 cm^−1^ are depolarized. 

In a Lu(ClO_4_)_3_ solution at 0.186 mol·L^−1^, the ν_1_(a_1_) ClO_4_^−^ band appears at 931.8 cm^−1^ (fwhh = 7.4 cm^−1^). In a concentrated solution (2.233 mol·L^−1^) the ν_1_(a_1_) ClO_4_^−^ band shifts to 934.2 cm^−1^ and broadens (fwhh = 12.0 cm^−1^). At the same time, the antisymmetric stretch, ν_3_(f_2_) ClO_4_^−^ shifts to slightly higher wavenumbers and broadens.

The Raman spectra of Lu(ClO_4_)_3_ (aq) reveal, in addition to the perchlorate-bands, weak bands in the low frequency region (50 to 400 cm^−1^) which are connected to Lu^3+^ (aq) species. In the isotropic scattering, a band appears at 396 cm^−1^ which does not exist in ClO_4_^−^ (aq). The band must stem from vibrations connected to the LuO_8_ skeleton. This isotropic band at 396 cm^−1^ has a symmetrical profile with fwhh at 50 cm^−1^. In the totally symmetric stretching mode of the LuO_8_ skeleton, the Lu^3+^ ion remains stationary while the oxygen atoms vibrate without changing the symmetry of the LuO_8_ unit. (It is strongly polarized with a depolarization degree at 0.005). An example for the Raman spectrum of a 0.186 mol·L^−1^ Lu(ClO_4_)_3_ solution is given in Figure 2. In the anisotropic scattering, broad and very weak bands appear at 113 cm^−1^, 161.6 cm^−1^, 231 cm^−1^, 261.3 cm^−1^ and 344 cm^−1^. Figure 3 shows the weak anisotropic modes for a concentrated Lu(ClO_4_)_3_ (aq) solution. Out of the seven theoretically predicted Raman modes only six could be observed. The second polarized LuO_8_ mode could not be observed. It is known that for hydrated metal ions in aqueous solution not all skeleton modes may be detected. In reality, the bands may be too broad and weak to be observed. The totally symmetric stretch, also called breathing mode of LuO_8_ is accompanied by a change in the polarizability ellipsoid but not in the dipole moment itself. This type of vibration which takes place with the conservation of all symmetry properties is thus called totally symmetric and has a depolarization degree ~ 0.

Replacing water with heavy water leads to an isotope shift to lower wavenumbers by a factor of ~0.948 in Lu(ClO_4_)_3_(D_2_O). The effect of deuteration on the LuO_8_ breathing mode was measured in Lu(ClO_4_)_3_- D_2_O solutions and a band at 376.5 cm^−1^ was observed (Appendix A). The theoretical shift of ν_1_ on deuteration (H_2_O/D_2_O considered as point masses) is given according to:ν_1_′ = ν_1_[m(H_2_O)/m(D_2_O)]^1/2^ = (396 cm^−1^) × 0.948 = 375.6 cm^−1^(6)
(The simple formula for calculating the isotopic shift of the ν_1_ mode is applicable because of the totally symmetric character of the normal mode of the LuO_8_ skeleton where Lu^3+^ remains stationary and only the oxygen atoms vibrate; the observed and theoretical depolarization degree is ~0.) 

The agreement between the measured ν_1_ symmetric stretch of the [Lu(D_2_O)_8_]^3+^ species and the calculated one is excellent. Furthermore, the DFT result calculated for [Lu(D_2_O)_8_]^3+^ with S_8_ symmetry imbedded in a PC leads to ν_1_ = 356.3 cm^−1^. If the DFT value at 376.3 cm^−1^ for [Lu(H_2_O)_8_]^3+^ is used to calculate the breathing mode for [Lu(D_2_O)_8_]^3+^, according to Equation (5), a value at 356.9 cm^−1^ follows. This agreement reinforces the assignment of this normal mode as a totally symmetric stretch with only insignificant contributions of the librations from heavy water.

Relative intensity measurements confirm that the scattering intensity of the ν_1_Lu-O mode is very weak with a scattering coefficient, S_h_ = 0.0024. The S_h_ values, defined as the R-corrected relative scattering efficiency of the M-O bands, were published for a variety of stretching modes of aqua metal ions in solution [27,28,29]. For Y^3+^, for instance, which resembles properties of the heavy rare earths, its ν_1_ breathing mode for [Y(H_2_O)_8_]^3+^ appears at 384 cm^−1^ [29] and also possesses a small relative intensity of 0.0025. Both ions, Lu^3+^ and Y^3+^, have a low polarizability and are classified as hard cations according to Pearson’s HSAB concept [63]. 

Perchlorate (Figure 2; Appendix A) was chosen as the counter ion because it is known as a weakly complex forming anion that does not substitute water in the first hydration sphere of the metal ions such as Lu^3+^. However, in a Lu(ClO_4_)_3_ (aq) at 2.233 mol·L^−1^, the mole ratio solute to water is 19.01. This water content is barely enough to completely hydrate the Lu^3+^ ion while the remaining 11.1 water molecules hydrate the three ClO_4_^−^ ions. In such a concentrated solution outer sphere ion pairs, [Lu(OH_2_)_8_]^3+^·ClO_4_^−^ form and this explains the slight concentration dependence of the peak position appearing at 394 cm^−1^ and the broadening of the band [27,28,29]. In a 0.189 mol·L^−1^ Lu(ClO_4_)_3_ solution, however, the ν_1_LuO_8_ band occurs as a symmetrical band at 396 cm^−1^ (fwhh at 50 cm^−1^). A hydrolysis effect which might have been the cause of the changes of these bands can be ruled out in these acidic solutions. In a recent La(ClO_4_)_3_ study on the influence of additional HClO_4_ in ternary solutions of La(ClO_4_)_3_ plus HClO_4_, measured in the low frequency region, the hydrolysis effect did not play a role at all [27,28,29].

To summarize: the Raman spectroscopy data clearly shows that in dilute Lu(ClO_4_)_3_ (aq) an isotropic mode appears at 396 cm^−1^ with fwhh = 50 cm^−1^ which represents the breathing mode of the LuO_8_ skeleton. Furthermore, five additional weak and broad bands, depolarized in character, could be found in concentrated Lu(ClO_4_)_3_ (aq). 

DFT frequencies and assignments of the LuO_8_ modes: The DFT frequencies for the LuO_8_ skeleton modes for [Lu(H_2_O)_8_]^3+^ imbedded in a polarizable dielectric continuum simulating the bulk water phase are given in Appendix A. The totally symmetric stretch of the LuO_8_ skeleton ν_1_ LuO_8_ gave the wavenumber position at 376.3 cm^−1^ in satisfactory agreement with the measured value (Appendix A). The remaining 6 Raman active modes are in fair agreement with the measured ones considering the simple model used. One theoretical mode could not be detected. Appendix A presents and describes the theoretical LuO_8_ skeleton modes for [Lu(H_2_O)_8_]^3+^ and its deuterated analog. Furthermore, the character of the ν_1_LuO_8_ mode as a totally symmetric stretch with a depolarization degree zero was also verified. In contrast to the frequency in the condensed phase, the breathing mode for the [Lu(OH_2_)_8_]^3+^ in the gas phase gave a frequency value at 347.1 cm^−1^. The totally symmetric LuO_8_ stretch is called the breathing mode because while vibrating, the geometry of the cluster does not change its shape, the symmetry remains and therefore this mode is not allowed in i.r. The DFT frequency for ν_1_LuO_8_ in vacuo is much smaller than the one calculated with a polarizable continuum because the latter method takes into account the influence of the bulk water molecules. It is important to estimate the accuracy of the DFT method and such a brief discussion is given in Appendix C.

The hypothetical nonahydrate [Lu(OH_2_)_9_]^3+^ in vacuo with its tricapped trigonal prism (TTP) structure gave a much lower value for the symmetric stretching mode namely at 329 cm^−1^ compared to 347.1 cm^−1^ for the in vacuo frequency of [Lu(OH_2_)_8_]^3+^. Furthermore, the Lu-O bond distances are much larger at 2.448 Å (three capping water molecules) and at 2.379 Å (6 prism water molecules). This is an additional and significant result for the existence of the octahydrate with its SAP structure. 

To summarize, embedding the [Lu(H_2_O)_8_]^3+^ species in a polarizable continuum, taking into account the effect of the bulk water, gave a reasonable agreement with the experimental frequencies found in Lu(ClO_4_)_3_ (aq). Appendix A shows the results of all theoretical LuO_8_ modes for [Lu(H_2_O)_8_]^3+^ and its deuterated analog. The DFT value for the Lu-O bond distance of the [Lu(H_2_O)_8_]^3+^ at 2.31 Å is in good agreement with the experimental structural data (Table 1).

### 3.3. Lu^3+^- Trifluorosulfonate in Aqueous Solution 

In Lu(CF_3_SO_3_)_3_ (aq), the Lu-O band is slightly overlapped with a polarized triflate band at 319 cm^−1^ and appears at 396.5 cm^−1^. The Raman spectrum is shown in Appendix A. A band separation resulted in two bands with the first band component at 319 cm^−1^ and the second band at 396.5 cm^−1^ (fwhh = 48 cm^−1^). The first band, a polarized band, stems from CF_3_SO_3_^−^ (aq) but the second, much weaker one, strongly polarized, represents the LuO_8_ breathing mode of [Lu(H_2_O)_8_]^3+^. The triflate in aqueous solution acts as a non-complexing anion and is suited, therefore, to study metal ion hydration. Band parameters and assignments of CF_3_SO_3_^−^ (aq) modes are given in ref. [29].

### 3.4. LuCl_3_ (aq) 

A Raman spectrum in R-format of two LuCl_3_ solutions, one at 3.199 mol·L^−1^ (R_w_ = 15.68) and a more dilute one at 0.478 mol·L^−1^ (R_w_ = 166) are presented in Figure 4A,B, respectively, in the wavenumber range from 50–1300 cm^−1^. The Lu-O stretching mode in the concentrated solution is down shifted and appears at 390 cm^−1^ and an additional very broad isotropic component at 205 cm^−1^ with a shoulder at ~254 cm^−1^ was observed. This finding is clear evidence that Cl^−^ substituted a water molecule in the first hydration shell of Lu^3+^. The second isotropic band is due to the restricted translation mode of water and water/Cl^−^ at 205 cm^−1^ (Figure 4A) while the broad band at 254 cm^−1^ appearing as a shoulder has been assigned to a Lu(OH_2_)_7_^3+^Cl^−^ stretching mode of a 1:1 Lu^3+^-chloro-complex, [Lu(OH_2_)_7_Cl]^2+^. The concentration of the 1:1 chloro-complex species is in equilibrium with the fully hydrated species, [Lu(OH_2_)_8_]^3+^. In contrast to concentrated LuCl_3_ solutions, the Lu-O stretching mode appears in dilute solutions (< 0.478 mol·L^−1^) at 396 cm^−1^ identical to the peak position in Lu(ClO_4_)_3_ (aq). A dilution series presented in Figure 5 shows the shift of the Lu-O stretching mode to higher wavenumbers from 390 cm^−1^ to 396 cm^−1^. In a 3.199 mol·L^−1^ solution with a mole ratio solute to water at 1 to 15.68, the peak appears at 390 cm^−1^, shifts to 394 cm^−1^ for a solution at 1.890 mol·L^−^^1^ (R_w_ = 28.51) and to 395 cm^−1^ for a 0.935 mol·L^−^^1^ (R_w_ = 59.00) solution and then remains constant at 396 cm^−1^ for solutions < 0.478 mol·L^−^^1^ (R_w_ = 115.99). This frequency shift is Geben Sie hier eine Formel ein. accompanied by a change of the fwhh which becomes smaller with dilution and in solutions < 0.478 mol·L^−1^ remains constant at 50 cm^−1^. This shows that with dilution the chloro-complex species disappears quickly and extrapolation of the Raman data show that in LuCl_3_ (aq) ≤ 0.5 mol·L^−1^ the Lu^3+^ cation is fully hydrated. Furthermore, the intensity of ν_1_LuO_8_ band as a function of concentration does not increase linearly in the LuCl_3_ solutions. However, the intensity increases less with concentration in contrast to the linear concentration dependence in Lu(ClO_4_)_3_ (aq). Such a linear increase in band intensity would be expected if the octahydrated Lu^3+^ species remained the only stable species. This shows, clearly, that a chloro–complex species must have formed in higher concentrated solutions at the expense of the fully hydrated Lu^3+^, [Lu(H_2_O)_8_]^3+^ ion (Figure 5). Quantitative Raman analysis revealed the species concentrations of fully hydrated Lu^3+^ (aq) and the 1:1 chloro-complex. The relationship between the integrated band intensity of ν_1_ LuO_8_, I_396_ and the solute concentration of Lu(ClO_4_)_3_ is a linear one (see Equation (1); Experimental Sect.). The measured integrated band intensity of LuO_8_ in LuCl_3_ (aq), I_396_, follows the given linear relationship between I_396_ and C_T_ established in Lu(ClO_4_)_3_ (aq) up to ~0.5 mol·L^−1^ but then levels off noticeably at higher LuCl_3_ concentrations (Appendix A). Obviously, above 0.5 mol·L^−1^ LuCl_3_ fractions of the fully hydrated Lu^3+^ (aq) are converted to a 1:1 Lu^3+^ chloro-complex species. The existence of higher chloro complexes than 1:1 can be convincingly ruled out taking into account the results of earlier anion exchange studies on aqueous rare earth chloride systems [37]. 

The mole fractions of both species are plotted in Figure 6. The fraction of the chloro-complex at 32%, in the most concentrated solution, is rather small and the fully hydrated species at 68% is still dominant. With dilution, the fraction of the chloro-complex species disappears quickly and at 0.5 mol·L^−^^1^ it has vanished. It should be pointed out that the chloro-complex species is the only species detectable by Raman spectroscopy in agreement with the majority of the solution chemistry studies put forward [31,32,33,34,35,36,37]. The formation of the 1:1 chloro-complex may be formulated according to Equation (7): [Lu(OH_2_)_8_]^3+^ + Cl^−^ ⇄ [Lu(OH_2_)_7_Cl]^2+^ + H_2_O (7)
and the formation constant for the 1:1 Lu^3+^- chloro-complex formation, K_1_, is according to Equation (8):(8)$$K_{1}=\frac{\left[{\mathrm{LuCl}}^{2+}\right]}{\left[{\mathrm{Lu}}^{3+}\right]\left[{\mathrm{Cl}}^{-}\right]}\times \frac{f_{{\mathrm{LuCl}}_{2}^{2+}}}{f_{{\mathrm{Lu}}^{3+}}\times \;f_{{\mathrm{Cl}}^{-}}}$$

33 and with K_1_’ the “concentration quotient” we get: (9)$$K_{1}=K_{1}^{\prime }\times \frac{f_{{\mathrm{LuCl}}_{2}^{2+}}}{f_{{\mathrm{Lu}}^{3+}}\times {\;f}_{{\mathrm{Cl}}^{-}}}.$$

The concentration quotient can be measured by Raman spectroscopy according to Equation (10):(10)$$K_{1}^{\prime }=\frac{\left(C_{T}-\left[{\mathrm{Lu}}^{3+}\right]\right)}{\left[{\mathrm{Lu}}^{3+}\right]\times \left[{\mathrm{Cl}}^{-}\right]}$$
where C_T_ is the total LuCl_3_ concentration and the concentrations in brackets denote the equilibrium concentrations of the fully hydrated Lu^3+^ and Cl^−^. The equilibrium concentration of Lu^3+^ can be measured by Raman according to Equation (1) and thus we obtain $K_{1}^{\prime }$.

The logK_1_′ value measured by Raman spectroscopy is −0.62 (I = 5.61 mol·L^−1^) and reveals the weak nature of the chloro-complex in LuCl_3_ (aq) at 23 °C. Literature values for logK_1_ for rare earth chlorides were compiled by Wood [31] and our logK_1_^′^ fits reasonably well with the literature data given in ref. [31] Applying the specific ion interaction theory [64] in order to extrapolate logK_1_^′^ to zero concentration, a logK_1_ value at ~ −1.3 is obtained. The nature of observed chloro-complex species has been controversially discussed. While NMR- and fluorescence spectroscopy [38,39,40,41,42,43] as well as the Raman effect detect chloro-complex species where chloride substitutes a water in the first hydration sphere, other methods such as ultrasound absorption [44,45,46,47] and DRS [30] may detect all 1:1 species, namely ion pairs where Cl^−^ is in the outer-sphere, in the outer-outer-sphere and also contact ion pairs/complex species. (DRS has however a considerable drawback in detecting complex species because symmetric complex species for which the dipole moment is zero cannot be probed.)

In a recent DRS study [30] the existence of outer-outer sphere ion pairs with two interposed water molecules was claimed to exist exclusively in LaCl_3_ solutions but no (direct ion pair) chloro-complex species could be detected. It is clear that different methods show different sensitivities probing species in aqueous solution. The sensitivity of DRS towards the different types of ion pairs is greatest for outer-outer sphere ion pairs less so for outer-sphere ion pairs and the least for complex species. Raman spectroscopy on the other hand fails to detect the outer-outer-sphere ion pairs but detects outer-sphere ion pairs and complex species/direct ion pairs. A Raman study taking into account the DRS data showed the different sensitivities for the detected species and this applies that different methods show different sensitivities toward the individual species [65]. For complete ion association models speciation data from different methods should be taken into account.

The weak chloro-complex species detected in LuCl_3_ (aq) could be also found in similar systems such as LaCl_3_ (aq), CeCl_3_ (aq) and YCl_3_ (aq) [27,28,29]. However, in AlCl_3_ solutions even at high concentrations, Cl^−^ does not substitute water of the first hydration shell of Al^3+^ [44,62] but forms outer-sphere ion pairs instead. The hydration shell of [Al(OH_2_)_6_]^3+^ is inert towards chloride substitution. Further experimental support for Lu^3+^- chloride complex formation stems from a recent THz FTIR study on a similar system, YbCl_3_ (aq) and YbBr_3_ (aq). While in YbCl_3_ (aq) chloro-complex species albeit weak are formed no such species exist in the bromide system [66].

To further verify chloro-complex formation in LuCl_3_ (aq), HCl was added. A ternary LuCl_3_-HCl solution composed of 1.589 mol·L^−1^ LuCl_3_ plus 6.047 mol·L^−1^ HCl was measured and from its isotropic scattering profile the isotropic scattering contribution of a 6.02 mol·L^−1^ HCl solution was subtracted. This difference spectrum reveals a band at 386 cm^−1^, a prominent and broad, slightly asymmetric component at 230 cm^−1^ and a small scattering contribution at 110 cm^−1^ (Figure 7). Clearly, chloro-complex species are formed in these solutions and the common ion effect results in the formation of a second chloro-complex, a 1:2 complex species in addition to the 1:1 complex. Results of anion exchange experiments of metal ions in aqueous HCl solutions including Lu^3+^ showed that lutetium has a negligible adsorption [31,37]. Therefore, higher Lu^3+^- chloro-complexes (complex anions) with *n* ≥ 3 can be ruled out. The formation of a second complex cation, the di-chloro-complex is formed according to Equation (11):[Lu(OH_2_)_7_Cl]^2+^ + Cl^−^ ⇄ [Lu(OH_2_)_6_Cl_2_]^+^ + H_2_O(11)

To summarize, a 1:1 chloro-complex species, [Lu(OH_2_)_7_Cl]^2+^ was detected in concentrated LuCl_3_ (aq) solution in equilibrium with the fully hydrated [Lu(OH_2_)_8_]^3+^ ion. The bands in the low frequency range assigned to the chloro-complex species could be characterized. A dilution series of LuCl_3_ solutions was studied by quantitative Raman spectroscopy and it could be shown that the chloro–complex dissociates completely to the [Lu(OH_2_)_8_]^3+^ (aq) and Cl^−^ (aq) (c < 0.5 mol·L^−1^). In other words the chloro-complex, [Lu(OH_2_)_7_Cl]^2+^, is very weak and dissociates rapidly with dilution (increasing water activity).

## 4. Conclusions

Raman spectra of aqueous Lu^3+^ perchlorate, triflate and chloride solutions were measured over a concentration range from 0.105 mol·L^−1^ to 3.199 mol·L^−1^. The weak, isotropic mode at 396 cm^−1^ (fwhh = 50 cm^−1^) was assigned to ν_1_ Lu-O of the LuO_8_ skeleton. In deuterated Lu(ClO_4_)_3_ solutions, a mode at 376 cm^−1^ was assigned to the breathing mode, ν_1_ Lu-O of [Lu(OD_2_)_8_]^3+^. In the anisotropic scattering of aqueous Lu(ClO_4_)_3_, five bands of very low intensity were observed at 113 cm^−1^, 161.6 cm^−1^, 231 cm^−1^, 261.3 cm^−1^ and 344 cm^−1^. Raman spectroscopic data suggest that perchlorate and triflate do not substitute water molecules in the first hydration sphere and the [Lu(OH_2_)_8_]^3+^ ion is stable in these solutions. Perchlorate and triflate are weakly complex forming anions. Although, no inner-sphere complex species could be detected in these solution, outer-sphere ion pairs of the type [Lu(OH_2_)_8_]^3+^·ClO_4_^−^ may be formed in concentrated Lu(ClO_4_)_3_ (aq). DFT frequency calculations of a [Lu(OH_2_)_8_]^3+^ imbedded in a polarizable dielectric continuum gave a ν_1_ Lu-O equal to 376.3 cm^−1^ in fair agreement with the experiment. The bond distances and angles of the [Lu(OH_2_)_8_]^3+^ imbedded in a polarizable dielectric continuum were also presented. The hydration enthalpy for Lu^3+^ (aq) could be simulated by DFT and ΔH_hyd(l)_ = −3847.7 kJ/mol was obtained in fair agreement with experimental values.

In LuCl_3_ solutions, in addition to the breathing mode, ν_1_ Lu-O of [Lu(OH_2_)_8_]^3+^, a second isotropic mode at 254 cm^−1^ was be verified. This new mode has been assigned to a Lu^3+^-chloro-complex species. A chloride ion, thereby, substitutes a water molecule of the first hydration sphere of Lu^3+^ (aq) forming a 1:1 chloro-complex. This is proof of the lability of the first hydration sphere of Lu^3+^ (aq). The Raman spectroscopic characterization of a 1:1 chloro-complex confirms recent results applying neutron- and X-ray scattering as well as EXAFS [12,20]. Quantitative Raman measurements allowed the determination of the fraction of the 1:1 chloro-complex and the fully hydrated species in concentrated LuCl_3_ solutions down to ~ 0.5 mol·L^−1^ where the chloro-complex vanishes.

## Figures and Tables

**Figure 1 molecules-23-03237-f001:**
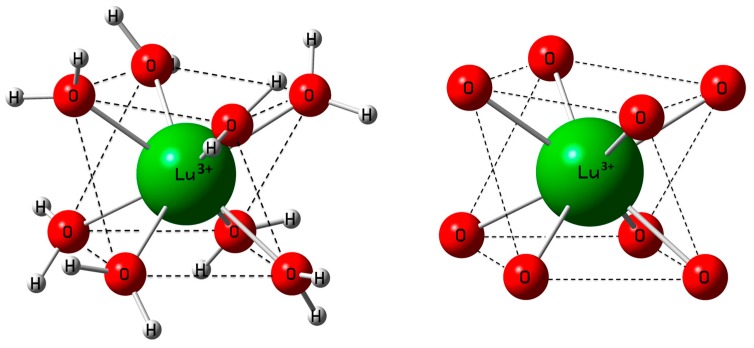
**Left**: Structure of the octaaqua Lu^3+^- ion (symmetry S_8_) as a gas phase cluster and imbedded in a polarizable dielectric continuum simulating the bulk water. At the **right**: The LuO_8_ skeleton (H_2_O as point masses) with its D_4d_ symmetry.

**Figure 2 molecules-23-03237-f002:**
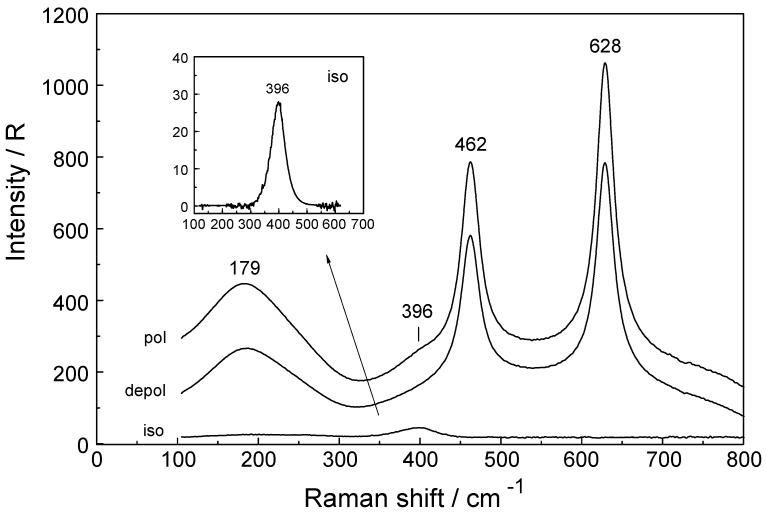
Raman spectrum in R-format (polarized, depolarized and isotropic scattering) of a 0.186 mol·L^−1^ Lu(ClO_4_)_3_ solution (R_w_ = 289.0). The isotropic band at 396 cm^−1^ is the symmetric stretching mode of the LuO_8_ skeleton of [Lu(OH_2_)_8_]^3+^. The ClO_4_^−^ (aq) deformation bands at 462 and 628 cm^−1^ are depolarized and do therefore not appear in the isotropic scattering. The broad band at 179 cm^−1^ is the restricted translation mode of water.

**Figure 3 molecules-23-03237-f003:**
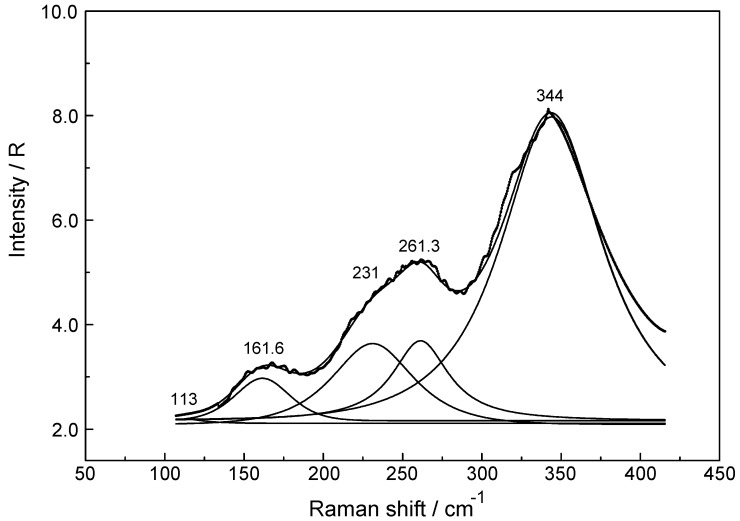
Anisotropic scattering profile in R-format of a 2.233 mol·L^−1^ Lu(ClO_4_)_3_ (aq) solution. Given are the measured curve, the sum curve and the 5 component bands of the band fit.

**Figure 4 molecules-23-03237-f004:**
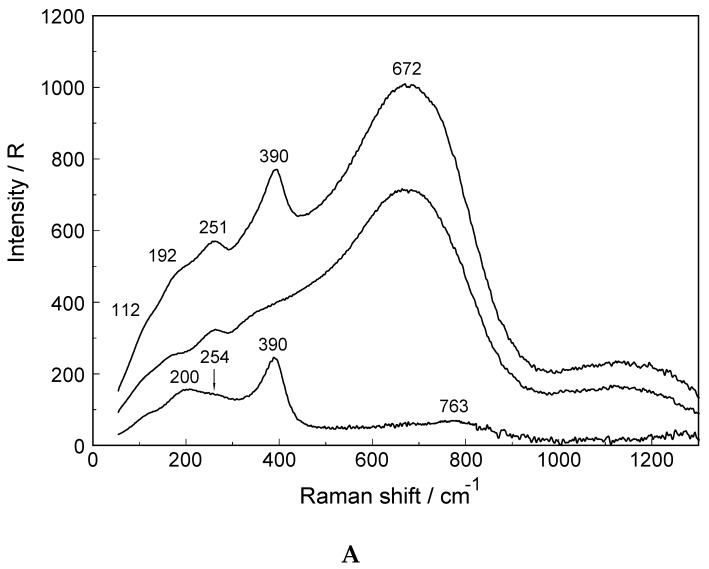
(**A**). Raman scattering profiles in R-format (from top to bottom: polarized, depolarized and isotropic scattering) of a 3.199 mol·L^−1^ LuCl_3_ solution. Note, the downshift of the Lu-O mode to 390 cm^−1^ compared to the one in Lu(ClO_4_)_3_ (aq) (compare Figure 2) is due to the substitution of Cl^−^ for a water molecule in the first hydration sphere, forming [Lu(OH_2_)_7_Cl_n_]^2+^. The extremely broad mode at 672 cm^−1^ is due to the librational water band influenced by the solute. The mode at 192 cm^−1^ is due to the restricted O-H···O band of H_2_O and the broad feature at ~251 cm^−1^ is assigned to [Lu(OH_2_)_7_Cl]^2+^. (**B**). Raman scattering profiles in R-format (from top to bottom: Polarized, depolarized and isotropic scattering) of a 0.478 mol·L^−1^ LuCl_3_ solution. Note, the ν_1_ Lu-O mode at 396 cm^−1^ for [Lu(OH_2_)_8_]^3+^. The extremely broad mode at 807 cm^−1^ in R_iso_ (R_pol_: 706 cm^−1^) is due to the librational water band influenced by the solute. The mode at 175 cm^−1^ (R_iso_) is due to the restricted translation O-H···O band of H_2_O.

**Figure 5 molecules-23-03237-f005:**
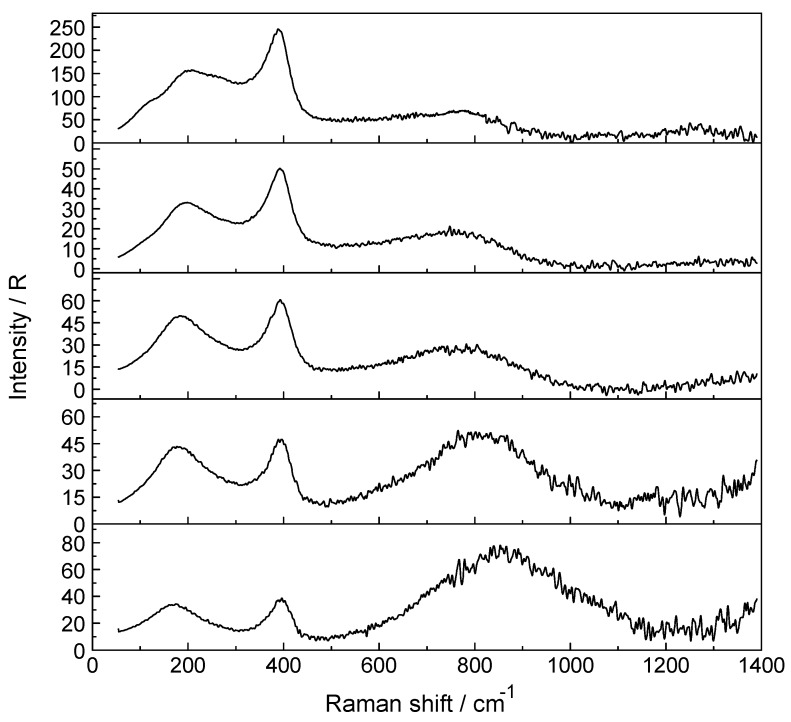
Isotropic Raman scattering profiles of aqueous LuCl_3_ solutions: from top to bottom: 3.199 mol·L^−1^ (R_w_ = 15.68), 1.890 mol·L^−1^ (R_w_ = 28.51), 0.935 mol·L^−1^ (R_w_ = 98), 0.478 mol·L^−1^ (R_w_ = 116) and 0.241 mol·L^−1^ (R_w_ = 229.86). The Lu-O mode in the most concentrated solution (3.199 mol·L^−1^) at 390 cm^−1^ compares to the one at 396 cm^−1^ in a 0.241 mol·L^−1^ LuCl_3_ (aq). This slight frequency shift is due to the substitution of Cl^−^ into the first hydration sphere, forming [Lu(OH_2_)_7_Cl]^2+^. The broad feature at ~254 cm^−1^ is assigned to [Lu(OH_2_)_7_Cl]^2+^. See also Figure 4A,B.

**Figure 6 molecules-23-03237-f006:**
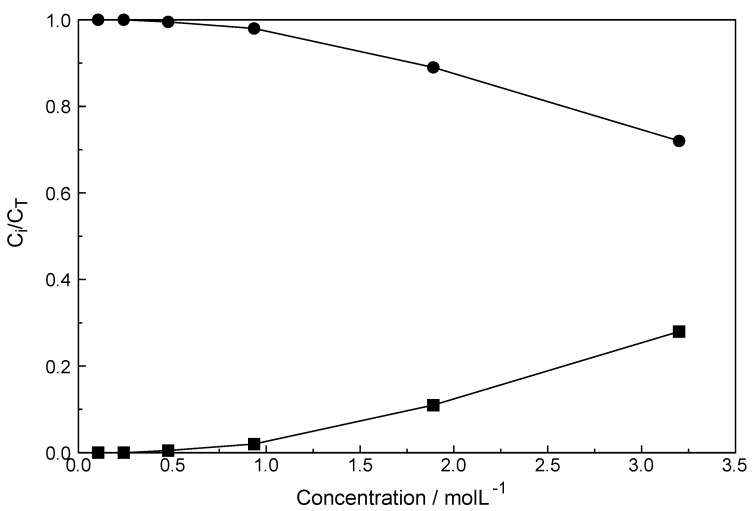
Fraction of species detected by quantitative Raman spectroscopy. The filled circles denote the [Lu(OH_2_)_8_]^3+^, the fully hydrated Lu^3+^ and the filled squares the 1:1 chloro-complex species.

**Figure 7 molecules-23-03237-f007:**
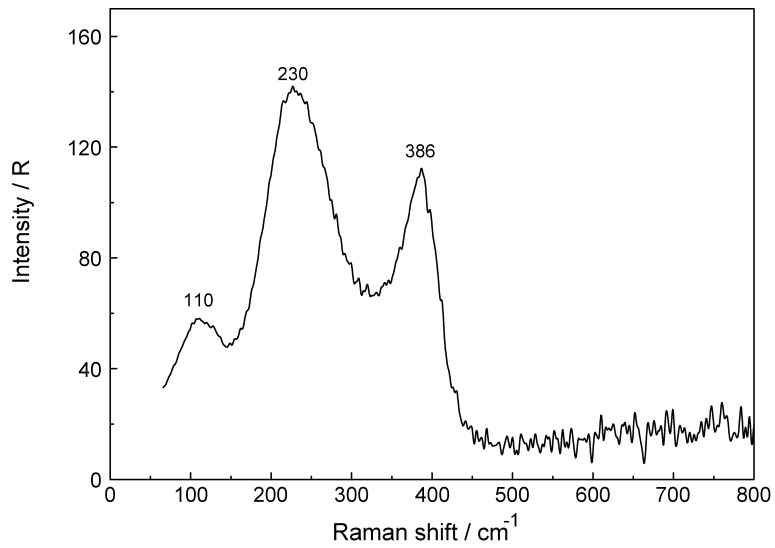
Isotropic Raman spectrum of a ternary LuCl_3_/HCl composed of 1.589 mol·L^−1^ LuCl_3_ plus 6.047 mol·L^−1^ HCl from which the isotropic scattering profile of a 6.02 mol·L^−1^ HCl solution was subtracted. The difference spectrum shows clearly that Cl^−^ must have substituted water molecules of the first hydration sphere of Lu^3+^. Lu-O mode of the complex, [Lu(OH_2_)_8−n_Cl_n_]^+3−n^ (*n* = 1, 2) and the broad mode at 230 cm^−1^ with a shoulder at 263 cm^−1^ and at 110 cm^−1^ are due to the chloro-complex species.

**Table 1 molecules-23-03237-t001:** Geometrical parameters such as bond distances and angles of [Lu(H_2_O)_8_]^3+^ imbedded in a polarizable dielectric continuum. Comparison of our DFT results (B3LYP/LANL2DZ) with published MD simulation and experimental results.

Bond Distances (Å) and Angels (°)	DFT Data/Gas Phase Cluster	DFT Data/Cluster + PC Model	ref. [25]	ref. [16]	ref. [24]	ref. [21]
Lu-O	2.350	2.311	2.338	2.307	2.37	2.32
O-H	0.981	0.982	-	-	-	-
H-O-H	109.29	110.45	-	-	-	-

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
