# Peer review of "Hydration and Ion Pair Formation in Aqueous Lu3+- Solution"

_molecules, 2018, doi:10.3390/molecules23123237_

Round 1

Reviewer 1 Report

This paper presents a study of the solution behaviour of the Lu3+ ion. The work is meticulously carried out and the conclusions are fully supported by the data.
I have no hesitation in recommending publication in Molecules, subject to the minor revisions listed below being done.

Page 5, lines 215-232
The discussion of the vibrational analysis is misleading. The authors split the 69 modes into two groups "water" and "LuO8". The LuO8 modes are the translational modes of the water molecules and that of the Lu3+ ion, while the water modes are the internal modes (2 x stretch and the bend) and the librational modes of water. Thus the statement in Line 220 that "The vibrations of the water modes, however, are decoupled from those of the LuO8 skeleton"is not correct, as the LuO8 skeletal modes are water translational modes. This must be corrected.

Lines 287-288
The authors should show the anisotropic spectra that exhibit the depolarized modes at  113 cm-1, 161.6 cm-1, 231 cm-1, 261.3 cm-1 and 344 cm-1.

Lines 394-398
Some elements of the equations have been translated into Chinese characters, there is a ? in eq. 8.

Author Response

Reply to the reviewer’s comments:

Reviewer 1:

This paper presents a study of the solution behaviour of the Lu3+ ion. The work is meticulously carried out and the conclusions are fully supported by the data.
I have no hesitation in recommending publication in Molecules, subject to the minor revisions listed below being done.

Page 5, lines 215-232
The discussion of the vibrational analysis is misleading. The authors split the 69 modes into two groups "water" and "LuO8". The LuO8 modes are the translational modes of the water molecules and that of the Lu3+ ion, while the water modes are the internal modes (2 x stretch and the bend) and the librational modes of water. Thus the statement in Line 220 that "The vibrations of the water modes, however, are decoupled from those of the LuO8 skeleton "is not correct, as the LuO8 skeletal modes are water translational modes. This must be corrected.

Lines 287-288
The authors should show the anisotropic spectra that exhibit the depolarized modes at  113 cm-1, 161.6 cm-1, 231 cm-1, 261.3 cm-1 and 344 cm-1.

Lines 394-398
Some elements of the equations have been translated into Chinese characters, there is a ? in eq. 8.

Reply:

1.       The symmetric stretching mode ν1 LuO8  is indeed a symmetric normal mode (depolarization degree = 0). The LuO8 normal modes may be nicely visualized in the viewing program of Gaussian. Again the normal mode considering the Lu-O symmetric stretch leaves the Lu3+ ion unchanged and the OH2 molecules vibrate as a whole. This is clearly the result of our Gaussian simulation on the Lu(OH2)8]3+ ion. We agree with the reviewer that the Lu-O modes may be frozen translations or rotations but it becomes clear that they are better termed Lu-O ligand modes. Furthermore, the Lu3+ solutions in D2O show the expected isotope shift according to eq. 6. The D2O vibrates also as a whole for this normal mode. A good discussion may be found in ref. 62 and especially the given literature therein. WE have enlarged the TableS1 and put in an explanation about the character of the modes.

2.       We have inserted a Figure (Figure 3) depicting the anisotropic scattering spectrum and the band fit of the profile and the component bands.

3.       I could not find these characters.

Reviewer 2 Report

The paper summarizes a series of measurements on model lutenium compounds; from experimental and simulated Raman spectra the structure of the metal and other properties are deduced. I agree that this is needed for better understanding of solutions containing this and other metal ions. The computations and experiments appear carefully conducted, the conclusions from the analysis of the data seem reasonable, and the data are clearly presented. I therefore recommend publication; perhaps the authors could consider some minor points before that:

in the introduction, they could mention similar attempts to decipher the structure of hydrated metals by Raman spectroscopy, e.g. J. Phys. Chem. B 2010, 114, 3574-3582

it is not clear which excitation wavelength was used for the presented spectra; can the Raman signal be clearly separated from eventual Lu3+ luminescence?

starting from line 145, the symbols in the equations should be somewhat more clearly defined/explained

it might be useful to discuss more the accuracy of the DFT computations

-      line 173, und -> under

Author Response

Reviewer 2:

The paper summarizes a series of measurements on model lutenium compounds; from experimental and simulated Raman spectra the structure of the metal and other properties are deduced. I agree that this is needed for better understanding of solutions containing this and other metal ions. The computations and experiments appear carefully conducted, the conclusions from the analysis of the data seem reasonable, and the data are clearly presented. I therefore recommend publication; perhaps the authors could consider some minor points before that:

in the introduction, they could mention similar attempts to decipher the structure of hydrated metals by Raman spectroscopy, e.g. J. Phys. Chem. B 2010, 114, 3574-3582

it is not clear which excitation wavelength was used for the presented spectra; can the Raman signal be clearly separated from eventual Lu3+ luminescence?

starting from line 145, the symbols in the equations should be somewhat more clearly defined/explained

it might be useful to discuss more the accuracy of the DFT computations

Reply

1.     The described data in the suggested journal are the ones about Mg2+ hydration. We have already published a paper about the Mg(II) hydration years ago (Rudolph et al., PCCP, 2003, 5, 5253–5261.) We believe the discussion of divalent ions is not the topic of this paper. We will, however, considering the quotation of the paper in a forthcoming publication dealing with Co(II), Ni(II), Cr(II) etc.

2.     The laser frequency used is given in the paper (4879.87 Angstroem). There are no excitation bands in the spectrum of Lu(III) and therefore, luminescence  cannot be excited. Lu3+ is not coloured.

3.     We defined and explained more clearly the symbols and notations used after line 144/145. We wrote the following:    The scattering geometries IVV=(X[ZZ]Y) and IVH=(X[ZX]Y) are defined as follows: the propagation (wave vector direction) of the exciting laser beam is in X direction and the propagation of the observed scattered light is in Y direction, the 90° geometry. The polarisation (electrical field vector) of the laser beam is fixed in Z direction (vertical) and the polarisation of the observed scattered light is observed in Z direction (vertical) for the IVV scattering geometry. For IVH the fixed electric field vector of the exciting laser beam in Z direction (vertical) and the observed scattering light is polarized in the X direction (horizontal). Thus, for the two scattering geometries it follows:   ……

4.     It might be useful to discuss more the accuracy of the DFT computations.

5.     We have incorporated a small paragraph as Appendix B in order to say something about this problem. This is indeed a good question. Too many times have theoretical models been used without questioning the theoretical approach. We believe, however, that it is not the scope of the present paper to discuss in detail the strength and limitations of the DFT models. We in cooperated the following small paragraph:   The application of the B3LYP method seems to be suitable to describing properties of the [Lu(OH2)8]3+ cluster reliably. For the cluster [Lu(OH2)8]3+ in the gas phase the totally symmetric stretching mode  ν1 LuO8 = 347.1 cm-1  and r0 = 2.350 Ã… were obtained.  Ab initio methods such as MP2 give similar results ν1 LuO8 = 356.5 cm-1 and r0  = 2.305 Ã… but 20 times more time consuming. With the HF method values 347.1 cm-1 and 2.376 Ã… are obtained. In all these cases the SDD basis set was used. Also other basis sets and density functionals were systematically investigated for hydrated Ln3+ clusters by A. E. Clark (J. Chem. Theory Comput. 4 (2008), 708). In order to obtain better agreement with experimental results it is necessary to consider the influence of the hydration shells beyond the octahydrate, [Lu(OH2)8]3+. Using B3LYP/SDD and the polarizable continuum model the deviation between the calculated and the experimentally observed totally symmetric stretching mode ν1 LuO8 amounts to 5% and 2 % for the geometry parameter r0. (Some shortcomings of the PCM model are mentioned in appendix A).